# Segmentation and Quantitative Analysis of Photoacoustic Imaging: A Review

**Thanh Dat Le [1]**, **Seong-Young Kwon [2]** and **Changho Lee [1,2,*]**

1   Department of Artificial Intelligence Convergence, Chonnam National University, Gwangju 61186, Korea; 208181@jnu.ac.kr
2   Department of Nuclear Medicine, Chonnam National University Medical School & Hwasun Hospital, Hwasun 58128, Korea; kwonsy@chonnam.ac.kr
*   Correspondence: ch31037@jnu.ac.kr; Tel.: +82-31-652-2779

**Abstract:** Photoacoustic imaging is an emerging biomedical imaging technique that combines optical contrast and ultrasound resolution to create unprecedented light absorption contrast in deep tissue. Thanks to its fusional imaging advantages, photoacoustic imaging can provide multiple structural and functional insights into biological tissues such as blood vasculatures and tumors and monitor the kinetic movements of hemoglobin and lipids. To better visualize and analyze the regions of interest, segmentation and quantitative analyses were used to extract several biological factors, such as the intensity level changes, diameter, and tortuosity of the tissues. Over the past 10 years, classical segmentation methods and advances in deep learning approaches have been utilized in research investigations. In this review, we provide a comprehensive review of segmentation and quantitative methods that have been developed to process photoacoustic imaging in preclinical and clinical experiments. We focus on the parametric reliability of quantitative analysis for semantic and instance-level segmentation. We also introduce the similarities and alternatives of deep learning models in qualitative measurements using classical segmentation methods for photoacoustic imaging.

**Keywords:** photoacoustic; quantitative; segmentation; application; deep learning

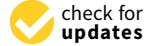



## 1. Introduction

Photoacoustic imaging (PAI) is a hybrid biomedical imaging technology using photoacoustic (PA) effect [1]. A non-ionizing pulsed laser is focused on biological tissues, which absorb the light energy, resulting in transient thermoelastic expansion. Rapid vibrations due to thermoelastic expansion generate broadband ultrasound waves. Using an ultrasonic transducer, the ultrasound waves are captured and analyzed with a PAI system to decode biological information. The advantages of PAI over other conventional optical imaging [2–5] include the dual contrast mechanism of the ultrasonic and optical imaging and the ability to perform multi-scale imaging. By optimizing the configuration of light excitation and ultrasound detection, PAI can delineate relative deep depth while maintaining high spatial resolution [6–14]. PAI reveals anatomical [13,15,16] and structural features, as well as endogenous molecules and conditions, such as melanoma, hemoglobin, collagen, and lipids, and physiological contrast features of intrinsic optical absorption [16–22], including blood flow, oxygen saturation, and metabolic rates. These advantages significantly contribute to successful preclinical and clinical studies and application [23–27].

PAI systems are divided into two types depending on their system configuration and performance (i.e., resolution and imaging depth) [1,11,28]: photoacoustic microscopy (PAM) [8,9,29–32] and photoacoustic tomography (PAT) [6,33–37]. PAM systems, depending on the laser or the acoustic focused beam configurations, can be used to visualize 1.2 mm-shallow microvasculature with a resolution of about several micrometers (optical-resolution/OR-PAM [31]), while deeper blood vessel mapping can be facilitated up to 3 mm (acoustic-resolution/AR-PAM [38]). By contrast, PAT does not use focused laser

or acoustic beam strategy and instead utilizes the universal back-projection algorithm (UBP) [39] or time-reversal (TR) [40] to make 2D cross-sectional and 3D volumetric data from multi-element US transducer arrays at a large acceptance angle within the field of view. By changing the setup of US transducer shapes and capturing design (e.g., planar, cylindrical, or spherical), several approaches integrating the temporal and spatial PAT data are available [1,41].

Quantitative photoacoustic imaging (QPAI) [42–45] aims to accurately estimate PA signals and physiological changes in tissue chromophores by measuring endogenous molecules (oxy/deoxyhemoglobin, melanin, collagen, lipids) or exogenous contrast agents. These are critical processes required to optimize the visualization of PAI. One of the first steps that broadly influence the QPAI accuracy is the segmentation process. Segmentation in PAI enables the visualization of structure [46,47] (diameter and tortuosity) and location based on similar attributes (blood vessel mapping [46], oxy/deoxyhemoglobin concentration [10], balloon catheter tracking [48], and lipid content [49–51]). Basically, mathematical methods were built to solve classical PAI segmentation, especially PAM [14,23,52] and PAT systems [33,37,53]. By mathematical theories, classic segmentation approaches were developed under several categories: the time domain (morphological segmentation methods), intensity domain (local/global thresholding segmentation methods), and frequency domain (wavelet segmentation methods). Furthermore, because of the simultaneous rapid developments of AI computing ability in this field, deep learning models have become the modern solution in many quantitative studies in PAI [54–56]. Based on classic segmentation approaches and quantitative analyses, supervised learning methods have been applied in current learning models, such as SegNet, FCN-8, or U-Net [57], to segment and analyze quantitative information with the same quality as manual tuning. Further, unsupervised learning methods can reveal hidden features in the domain, with unique advantages in resolving current problems and promoting the clinical application of developed methods. One of the key advantages of deep learning is that it enables low-cost and optimal PA system performance.

In this review, we summarize the current segmentation and quantitative analysis in PAI. In Section 2.1, we briefly introduce the classical approaches and techniques of segmentation utilized in the quantitative analysis of PAI. Deep learning network architectures are presented in Section 2.2. Finally, the findings are discussed and summarized in Section 3.

## 2. Segmentation and Quantification for PAI

### 2.1. Classic Segmentation and Quantification Approaches

Since the early days of computation system evolution, segmentation and quantification have made an important contribution to the analysis and measurement of medical image data, including the biometric index [58,59] and visualization of body parts [60,61]). Based on conventional medical imaging modalities [60–62], (CT [63,64], X-ray [65,66], MRI [64,67–69], PET [70–72], and ultrasound [73–75]), segmentation and quantification have been proposed and challenged frequently in response to improvements in imaging techniques. By validating the correlation and properties of the image, classic methods were proposed as basic concepts. Especially in PAI, based on the correlation and properties of digital ultrasound signals, we reviewed classical segmentation and quantification approaches to solve basic image processing challenges: thresholding segmentation, morphology segmentation, and wavelet segmentation. Table 1 lists the classic PAI approaches, and processes developed to date.

**Table 1.** Classic segmentation and quantification in PAI.

| Type | Description | Advantages | Disadvantages | Applications | Paper(s) |
|---|---|---|---|---|---|
| Thresholding segmentation | Based on a histogram of image intensity to detect bias | (1) Manual/adaptive flexibility (2) Simplest process | (1) Manual process (2) Cut-off weak signal mixed with noise | Leveling breast cancer Tissue chromophore measuring Micro-vessel mapping | [76–79] |
| Morphology segmentation | Based on the partitioning of image intensity into homogeneous regions or clusters | (1) Boundary region to separate background/target (2) Measurement of shape | (1) Complex algorithms (expensive computation systems) | Quantitative measurement of oxygenation Tracking/monitoring of vasculature Three-dimensional mapping | [46,47,80–82] |
| Wavelet-based frequency segmentation | Based on the frequency domain to determine the difference in absorption coefficient | (1) Consistent with the characteristics of wide-band ultrasound transducers (2) Measurement of quantitative coefficient | (1) Complex algorithms (expensive computation systems) | Blood vessel segmentation Enhanced visualization of biological cells | [83–85] |

### 2.1.1. Thresholding Segmentation

Thresholding-based segmentation [86] is the basic method and tool available for PAI segmentation. Using simple differential comparison based on signal amplitude, regions of interest (ROI) are segmented out of the background by selecting the average threshold value $T_{offset}$. In Equation (1), the values of binary image $I_{thres}$ are segmented into object (as Equation (1) or background (as 0) when values of a gray-level image $I_{val}$ are compared with the threshold value $T_{offset}$.

$$I_{thres}(x,y) = \begin{cases} 1 & | \ I_{val}(x,y) \geq T_{offset} \\ 0 & | \ I_{val}(x,y) < T_{offset} \end{cases} \tag{1}$$

In general, local threshold segmentation is almost inefficient because of the operator dependence as well as the single channel used [87]. Therefore, to automate the thresholding process, several global thresholding methods were used as an additional step for segmentation and quantification. The tumor edge could be segmented by the threshold segmentation algorithm. Zhang et al. [77] combined it with dynamic programming to define the breast cancer types based on color differences between the tumor mass (blue boundary) and the normal skin surface in PAI images (Figure 1a) into 06 classes based on the foreground threshold (FT) from the biopsy-proven negative category. The boundary of breast tumors could be used as the ground truth in learning-based segmentation. In another application, Khodaverdi et al. [78] applied automatic threshold selection (ATS) to support the adaptive matched filter (AMF), which classified different tissue chromophores in malignant melanoma (red color) on the ultrasound images (Figure 1b). The target of ATS was to identify an optimized threshold value based on the difference between the size of connected components [88] and the size of the largest connected components [89]. After identifying the highest abrupt changes connected with component fitting [90], the selected threshold was chosen from the estimated highest abrupt change in the number of pixels and depicted as the red area in Figure 1(bi). Conversely, Raumonen et al. [79] selected the changes in the relative proportion of voxels with different threshold values (follow 1/x-function) to select the quantitative threshold ranges (Figure 1(ci)). Using different filtering thresholds, the images with the maximum intensity projections (MAPs) showed the details of root (Figure 1(cii)) and an increased number of the small branches (from Figure 1(ciii–cv)). Otsu et al. [91] introduced a method to select a locally maximizing variance between classes. Sun et al. [82] showed the automatically estimated local offset value based on the histogram of the full three-dimensional OR-PAM data into multiple sub-parts for segmentation. Due to the limited intensity of noise interference that can

lead to erroneous segmentation, both approaches suppressed the background noise via a 3D Hessian mask and high-frequency enhancement. The intensity transformation was used to distinguish between blood vessels and background (intensity-based segmentation). Another study used Otsu's segmentation by Cao et al., 2018 [92] to segment micro-blood vessels from PAM to track the hemodynamic events in ischemic stroke.

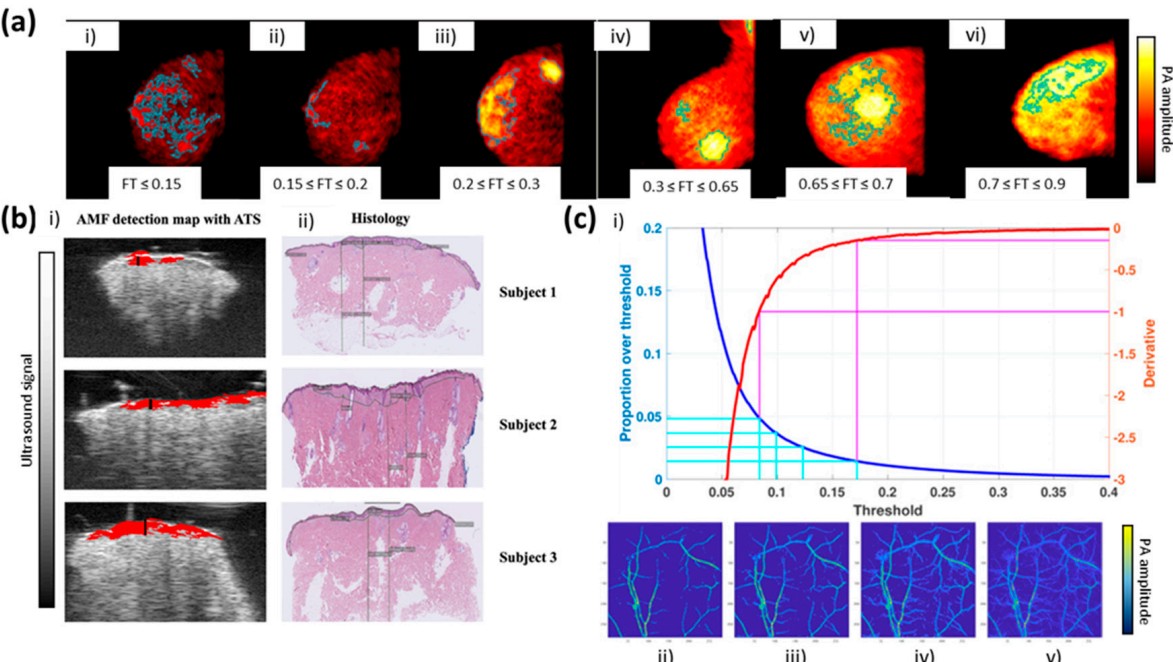

**Figure 1.** (**a**) BI-RADS 1–6 levels based on leveling tumor mass by foreground threshold (FT) values (blue boundary) and breast skin layer: (i) negative (FT ≤ 0.15); (ii) benign (0.15 ≤ FT ≤ 0.2); (iii) probably benign (0.2 ≤ FT ≤ 0.3); (iv) suspicious (0.3 ≤ FT ≤ 0.65); (v) highly suggestive of malignancy (0.65 ≤ FT ≤ 0.7); (vi) known biopsy-proven (0.7 ≤ FT ≤ 0.9). Reproduced with permission from Zhang et al. [77] and published by IEEE, 2018. (**b**) Adaptive matched filter (AMF) detection based on automatic threshold selection (ATS) in malignant melanoma samples (red color) using ultrasound images (i), compared with histological details showing the growing of tumors (ii). Reproduced with permission from Khodaverdi et al. [78]; published by OSA, 2021. (**c**) By selecting the 1/x-curve function and the very slow decrease in the proportion over the threshold (blue curve), the threshold filtering level can be selected quantitatively (i). Decreased limit threshold values (magenta) based on the proportion over the threshold (red); the maximum-intensity projections are shown from (ii) to (v). Reproduced with permission from Raumonen et al. [79]; published by OSA, 2018.

### 2.1.2. Morphology Segmentation

Morphology segmentation [93] (or homogeneous region segmentation) was used to measure the variance of regional homogeneity and the boundary between background and target (diameter, density, and tortuosity). In the MAP image, the basic method is to apply the gradient operator to determine the rapidly changing high and low values to determine the biological targets in PAI based on the closed curve. The local behavior of the second-order gradient projection based on n-dimension Hessian matrix was used to describe the boundaries of biological tissues highlighting the background and other tissues differentially. Mathematically, the eigenvalues of the 2D Hessian matrix represented long-twist targets (blood vessels) in the MAP image. A feature map was defined in Figure 2(ai) under the conditions of $|\lambda_1| \leq \varepsilon$ and $\lambda_2 < 0$ ($\varepsilon$ is a small fraction) in Equation (2) [94].

$$f = \max_{s_{min} \leq s \leq s_{max}} \begin{cases} 0, & \{\lambda_2 > 0\} \vee \{|\lambda_1| > \varepsilon\} \\ |\lambda_2| & \end{cases} \tag{2}$$

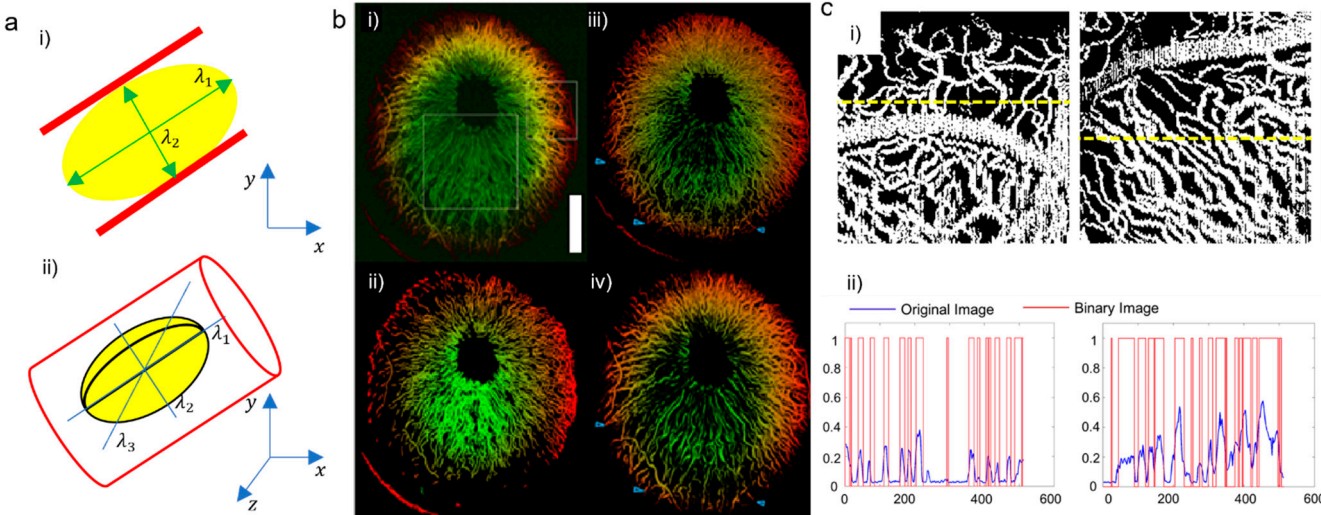

**Figure 2.** (**a**) Vessel boundary (red) information obtained using the multiscale Hessian-based segmentation algorithm in (i) 2D ($\lambda_1, \lambda_2$) image and (ii) 3D ($\lambda_1, \lambda_2, \lambda_3$) volume; (**b**) the rat's iris in (i) the original photoacoustic microangiography, (ii) Hessian segmentation using the 2D MAP image, (iii) 3D volume results of clinical CT-Hessian segmentation, and (iv) 3D volume results of Hessian segmentation using PAM-modified algorithm; (**c**) comparison of (i) the intensity with (ii) the extracted vessel profile. Reproduced with permission from Zhao et al. [81]; published by *J. Biomed. Opt.*, 2018.

The eigenvalues of the 3D Hessian matrix representing vessels are indicated in Figure 2(aii). Feature volume was defined under the conditions of $|\lambda_1| \leq \varepsilon \ll |\lambda_2|$ and $\lambda_2 \approx \lambda_3 < 0$ ($\varepsilon$ is small fraction) in Equation (3) [94].

$$
f = \max_{s=[s_{min}, s_{max}]} \left\{ \begin{array}{l} 0, \quad \{\lambda_2 > 0\} \vee \{\lambda_3 > 0\} \vee \{|\lambda_1| > \varepsilon\} \\ \left| \left[ 1 - \exp\left(-\alpha \frac{|\lambda_2^2|}{|\lambda_3^2|}\right) \right] \left[ \exp\left(-\beta \frac{|\lambda_1^2|}{|\lambda_2\lambda_3|}\right) \right] \left[ 1 - \exp\left(-\gamma(\lambda_1^2 + \lambda_2^2 + \lambda_3^2)\right) \right] \right| \end{array} \right. \tag{3}
$$

where $\alpha, \beta, \gamma$ denotes threshold multipliers that control the sensitivity of the suppression of noisy, plate-like, and blob-like structures. After mapping, the intensity transformation image *g* was measured to fit the enhanced feature map (2D) or feature volume (3D), as shown in Equation (4), with adjustable parameters representing the critical intensity *m* and the scaling factor *k*.

$$
g = \frac{1}{1 + \left(\frac{m}{f}\right)^k} \tag{4}
$$

In the last step, the growing region was expressed as pixels, with the maximum intensity value in the enhanced images seeded from regional growth's center, and the vessels were segmented to binary areas (Figure 2(ci)). The distance metric (DM) was defined as the actual path length of a vessel by the linear distance, whereas the inflection count metric (ICM) was defined as the length of a curve's inflection points and the DM. The sum of angles metric (SOAM) was summed along a curve's path length. All three product parameters of vessels were measured from the binary images [95,96]. Yang et al. [80] first introduced multi-parametric quantitative microvascular imaging using a 2D-Hessian matrix in an OR-PAM system. The two-step marching was used to generate centerlines of vascular and four types of morphological parameters (diameter, density, tortuosity, and fractal dimension) on multiple in 3D mouse scanning OR-PAM images. By developing a 3D vascular boundary, Zhao et al. [81] introduced the vascular information quantification algorithm for application in rat iris (Figure 2(biv)), and compared the 2D vascular boundary (Figure 3(bii)) with a 3D vascular application in clinical CT (Figure 2(biii)). The 3D vascular information quantification algorithm fixed the problem and enhanced the diagnostic capability of the OR-PAM system for vascular-related applications in vivo, but only the

MAP image is shown (Figure 2c). Sun et al. [82] applied a similar 3D vascular informa-tion algorithm in a full 3D framework. Compared with the 2D algorithm, the structural parameters obtained using the 3D vascular boundary algorithm were significantly closer to reality. Mai et al. [46,47] used a 2D Hessian matrix for the label-free quantitative analysis of three-dimensional OR-PAM and monitoring carfilzomib solution injection in the MAP images obtained via in vivo experiments.

### 2.1.3. Wavelet-Based Frequency Segmentation

Frequency-domain PAI via transformation from time-domain PAI showed different biological optical absorption coefficients in waveform bandwidths [97]. "Spectral unmix-ing" [98] refers to the process of decomposing the spectral signature of a mixed A-scan into a set of separate bands. In PAI, most transducers were the high-bandwidth type (piezoelectric transducers) [99]. Because the acoustic signal was released at a 360° angle, the transducers received the primary signal from the target accurately, with the second signal derived from uninterested targets with a similar absorption coefficient and could not be simply segmented in the time-domain.

Cao et al. [84] separated crystalline cholesterol from cholesteryl ester in an intact artery to diagnose cardiovascular disease via spectral unmixing with k-mean clustering [100]. The main procedure highlighted in Figure 3(ai), based on raw data, reveals pre-processing of the B-scanned image via bandpass filtering and Hilbert transformation for each A-scan. The Gaussian sliding window is moved gradually to remove the spectral leakage, followed by conversion to the frequency domain via discrete Fourier transform (DFT). After calibrating the sensitivity, the frequency domain data matrix was created following the normalization of the component in the selected bandwidth. The final form of the frequency-domain matrix was presented with each row representing all the pixels, while each column showed the tailored spectra for each sliding window. By calibrating the spectral matrix with k-mean clustering, the composition array was reshaped. Figure 3(aii) shows the traditional UST with PA intensity images with cholesteryl ester and cholesterol (green) upper background noise (white). For a more efficient measure, the denoised PA signal (Figure 3(aiii)) was applied. In Figure 3(av), the white color represents cholesteryl ester and segmentation in purple color, which denotes cholesterol. The accuracy of this phantom was 98.4%. The obtained maps of the slope, y-intercept, and mid-band fit distribution are displayed in Figure 3(av–avii), showing the confirmed difference, as well as spectral distribution using the K-mean clustering method.

Using different scale-specific contrast in the frequency domain, Moore et al. [85] introduced the F-Mode to reconstruct raw RF data in the PAI system to measure the differentiation between RF lines. In Figure 3(bi), each RF line as frequency domain was processed by the fast Fourier transform (FFT). The number of bands $K$ in the FFT was calculated based on the sampling frequency $f_s$ and the desired resolution of resultant frequency domain spectrum $\Delta f$ (Equation (5)).

$$K = f_s / \Delta f \tag{5}$$

The corresponding power spectra were partitioned into Q discrete frequency bands, it applied the F-Mode to each RF step in the generated map, which may be assumed to be on the same scale as the MAP image. The actual magnitude of a pixel's summed power with respect to all other pixels in the same band defines its intensity in an F-Mode image. Each F-Mode image presents a separate dynamic range, with a signal-to-noise ratio (SNR) resistant to substantial fluctuations in transducer sensitivity. Pixels derived from an image carrying a minimum in each frequency band may be indistinguishable from the background in each F-Mode image, thus masking the object while significantly increasing the visibility of the remaining features. However, because of the shifting spectra (Figure 3(ai)), these same pixels may have the highest overall value in an image. For example, in vivo imaging with a modular PAM system was used to capture the zebrafish larvae for classification based on the three main trunk vessels: 5 day-post (fertilization): dorsal aorta (DA), posterior cardinal

vein (PCV), and intersegmental vessels (ISV). As shown in Figure 3(bii), the MAP revealed that all the three trunk vessels had similar highest frequency-domain intensity and therefore could not be segmented according to the threshold value or Hessian filter because they connected directly with each other (PCV with ISV). By selecting different frequency bands centered at 12.5 MHz (Figure 3(biii)), 31.5 MHz (Figure 3(biv)), and 97.5 MHz (Figure 3(bv)), each trunk vessel was shown or removed. Especially in this case, the 97.5 MHz frequency band delineated the ISV and its connection with PCV.

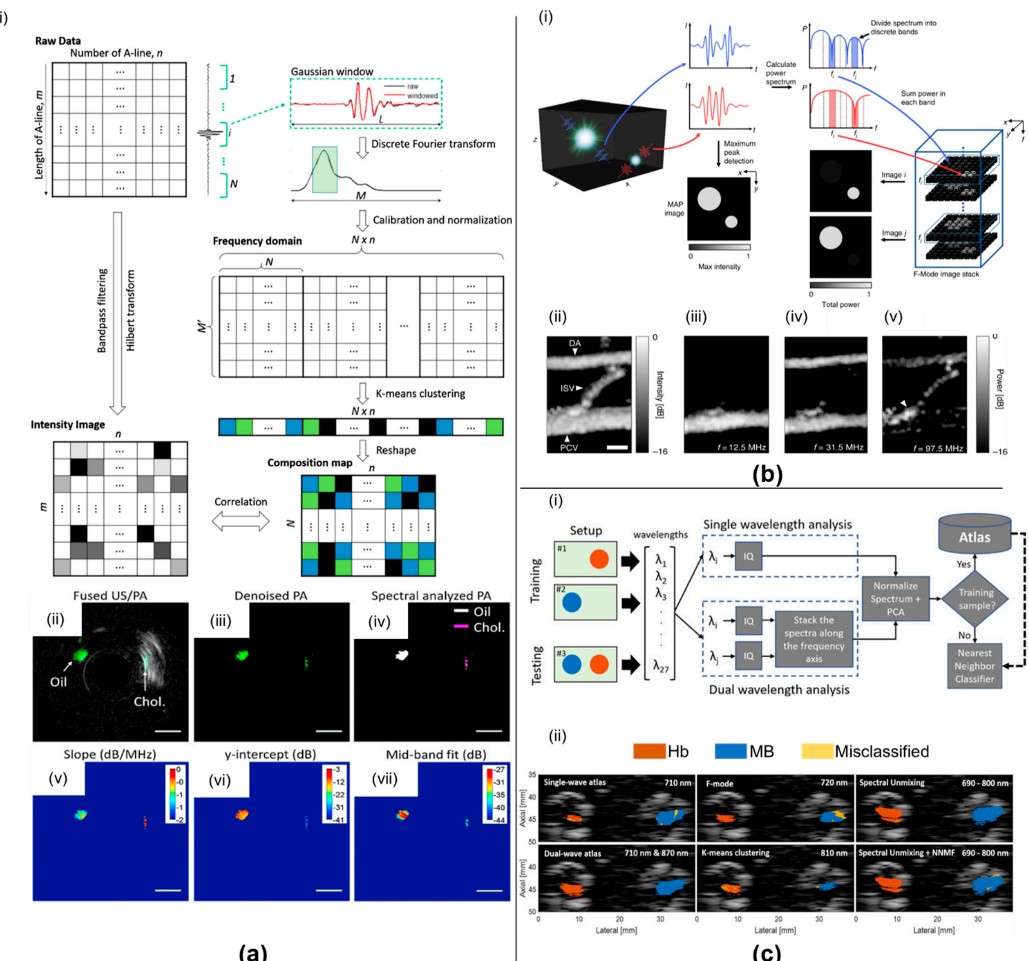

**Figure 3.** (**a**) Spectral analysis workload of photoacoustic signal; (i) the depth m and the scanning length n; by cropping Gaussian window L, the frequency domain after calibration was presented using the tailored spectrum of Gaussian matrix; using a shorter K-means clustering, the composition map was applied to B-scan; (ii) the PAI (gray) and derived composition spectral map (green); (iii) PAI based on signal thresholding; (iv) reconstructed composition map, marked by cholesterol (purple) and cholesteryl ester (white); (from v to vii) spectral maps showing slope, y-intercept, and mid-band fit. Reproduced with permission from Cao et al. [84]; published by Elsevier, 2017; (**b**) the overview of the F-Mode technique; (i) two phantoms were stimulated and divided into discrete frequency spectra; the reconstruction was segmented in the MAP image stack; (ii) the application of F-mode for scale-specific live-visualization of zebrafish vessels contains: the dorsal aorta (DA)/intersegmental vessel (ISV)/posterior cardinal vein (PCV) (iii) only PCV; (iv) removed ISV; (v) small ISV hidden under PCV. Reproduced with permission from Moore et al. [85]; published by *Nature*, 2019; (**c**) (i) overview of the differentiate photoacoustic signal sources by proposed method, (ii) separated ROIs for methylene blue (MB) and hemoglobin (Hb); (ii) ROIs of MB and Hb using different approaches. Reproduced with permission from Gonzalez et al. [83]; published by *Frontiers*, 2021.

Both methods mentioned above had certain requirements, which rendered them difficult to use in feasibly responding to most applied systems [84,85]. Because of the need to label the regions for each desired target, and these labeled regions relied on a priori information with limited number and location of materials for differentiation. Gonzalez et al. [83] proposed a novel acoustic-focusing frequency analysis to distinguish the changing of material photoacoustic responses. The proposed method utilized a classification framework using a dictionary of sets containing specific photoacoustic-sensitive materials (in Figure 3(ci)). The sources of intelligent classification were prepared from the set of frames averaged with coherence masks of the regions of interest. The feature extraction was described based on the amplitude and frequency, as well as the coefficient of principal components, to determine the atlas of photoacoustic-sensitive materials. As shown in Figure 3(cii), the comparison with K-mean clustering by Cao [84] or F-Mode by J. Moore [85] showed a disadvantage because of the improper detection of signals from methylene blue (MB) or blood (Hb). The dual-wavelength atlas showed the strongest sensitivity (0.88) with traditional spectral unmixing (0.89), higher than K-mean clustering (0.7) and F-mode (0.63). Due to the highest specificity and accuracy (0.87–0.88), the classification framework overcame the K-mean clustering specificity (0.67–0.64) and F-mode (0.67–0.62) and was higher than traditional spectral unmixing (0.81–0.86).

## 2.2. Deep Learning Segmentation Approaches

### 2.2.1. Overview of Deep Learning Segmentation Networks

Among the various benchmarks for image analysis, deep learning rapidly outperformed standard technologies, improving medical imaging analysis [101,102]. AlexNet was the first convolution neural network (CNN) winning against a support vector machine (SVM) at the ImageNet Large-Scale Visual Recognition Challenge 2012 (ILSVRC [103]). Since then, it has been developed into modified and diverse architectures [104] for application in most of the current medical imaging solutions [57,105–108]. Deep learning was also utilized in PAI as an innovative and economical solution. Table 2 lists the deep learning applications used in PAI.

**Table 2.** Deep learning projects facilitating PAI segmentation and quantification.

| Type | Description | Advantages | Disadvantages | Application | Paper(s) |
|------|-------------|------------|---------------|-------------|----------|
| Supervised learning | Based on label observation in the training prediction model | (1) Classification and regression (2) Simplest learning process | (1) Manual process (2) Limited by current knowledge | • Analysis of oxy-deoxy hemoglobin levels in blood • Classification of breast cancer | [56,77,109,110] |
| Unsupervised learning | Desired decisions without specific or explicit sample instruction | (1) Clustering/anomaly detection without instruction. (2) Quantitative measurement with hidden features | (1) Poor accuracy | • Blood vessel concentration • Blood vessel segmentation | [111–113] |

### 2.2.2. Supervised Learning Segmentation

The first important application of supervised deep learning is in labeling dataset. The well-organized dataset may be used to identify and determine targets in the learning process, which is desirable in reducing the training time or diversifying the learning model by shortening or enriching the meaning of the data set using image processing techniques. In PAT, the 2D MAP image processing tasks were used after the forward back-projection (FBP) to reconstruct biological images based on physical information (diameter, size, and angle) and labeling. Zhang et al. [77] and Ma et al. [109] used deep learning techniques to classify and segment breast cancer using two different methods: SVM and U-Net. With the helping of SVM, Jiayao successfully classified the reconstructed photoacoustic breast cancer images into six classes that were used in threshold segmentation (Figure 1a). To improve the segmentation process, Ma used the U-Net method to extract fibro-glandular tissues of breast, skin, and fat. The tumor, which was not classified or detected using the same features as notation targets, would be detected as an anomaly.

Chlis et al. [110] introduced the sparse-U-Net (S-UNET) for automatic vascular segmentation in multispectral optoacoustic tomography (MSOT [114]) to resolve oxygen consistence levels in the flowing blood. As shown in Figure 4(ai), the recorded MSOT stacks were transformed from stacks of 28 wavelengths to a probability map (wavelength combination) that corresponded to a ground-truth binary mask at wavelengths 810 nm (THb) and 850 nm (HbO$_2$), based on consensus between two clinical experts. The L1 regularization [115] was used with an additional term $\lambda|\beta|$ ($\lambda = 0.01$ was a scalar hyperparameter and beta was the first $1 \times 1$ convolutional layer) to eliminate unnecessary features of noise and background, while ensuring the target blood signals were measured. Using non-negative weights, a few relevant wavelengths were assigned to positive weights, while the rest were set to zero and executed in the model. The rest of the segmentation process was the original biomedical U-Net [116]. By selecting the dataset of 164 raw MSOT images, the S-UNET successfully showed the human vasculature segmentation from MSOT images. As shown in Figure 4(aii), using different vascular diameters ranging from the size of the cephalic vein to the radial artery compared with the 850 nm channel of the MSOT image (input image) and ground truth mask (blue in the true mask), the predicted results (red in predicted mask) showed a similar overlap (difference) and Dice coefficient up to 0.86 (higher than original U-Net with 0.75).

Ly et al. [56] improved B-scan segmentation in a dual-fast scanning photoacoustic microscopy system in vivo. The CNNs were used in B-scan segmentation for profiling shallow depth vessels in 3D volumetric data. The sliding window architecture (Figure 4(bi)) showed the transformation from a high-resolution B-scan to $256 \times 256$ patches. The deep learning models (U-Net/SegNet-5/FCN-8) were used to predict patches and recover the full B-scan original size. Each prediction (skin/vessels) was multiplied with each B-scan to determine the segmentation of skin and vessels (Figure 4(bii)). Using 4800 B-scan pre-treatments, patches were extracted from each B-scan and added to the U-Net/SegNet-5/FCN-8 to compare the performance. The U-Net showed the best performance with 99.53% accuracy and 75.29% BF-score (Boundary F1 contour matching score). The visualization in 3D volumetric of the human palm (Figure 4(ci)) and foot (Figure 4(cii)) showed good performance segmentation (Figure 4(ciii)—human palm/Figure 4(civ)—foot) in real-time scanning using the trained slide-U-Net algorithm (Figure 4c). Based on the coronal plane (Figure 4(cv,cvii)) and sagittal plane (Figure 4(cvi,cviii)), the vessels of the human palm showed the possibility of enhancing and visualizing a map of the skin surface and underlying vasculature.

### 2.2.3. Unsupervised Learning Segmentation

Until now, the performance and effectiveness of supervised and unsupervised segmentation methods were compared via surveys. Multiple quantification of endogenous chromophores (hemoglobin, melanin, and lipids) in supervised learning segmentation requires the classification of the labeling process to identify the correct label based on the known patterns (background). To identify unknown rules, unsupervised learning methods involved clustering or groups based on training samples. Yuan et al. [111] extracted the vessel images from the OR-PAM system and compared traditional K-mean segmentation methods with deep learning models as a fully convolution neural network (FCN) and U-Net. As described in Figure 5(ai), the FCN model configured the size of the convolutional kernel to $3 \times 3$ with a stride of 2, as expected, to preserve the most useful features of the vessel in PAM images. In the deconvolution process, a final operation with a stride of 8 was performed to eliminate unnecessary features (short line, noisy signal, . . . ) and maintain the regression information (long boundaries, clear edge, . . . ). In the same experiment, U-Net was used in a robust and wide application field as FCN. Based on subtle differences, the U-Net combined the low-level features of the decoding portion to avoid feature loss caused by the pooling layers in the network when the size of the convolutional and deconvolutional kernel was maintained at $3 \times 3$ with a stride of 2. However, both FCN and U-Net show characteristic limitations of models regardless of different hyperparameter tunning and an increasing number of iterations or training set size. By combining FCN and U-Net,

Hy-Net was created to optimize the results of the two models by avoiding the uniqueness of the output. The need for additional GPU memory in Hy-Net (77.56 MB) than in FCN (19.07 MB) and-U-Net (58.44 MB) increased the number of parameters, whereas Hy-Net showed a better performance than FCN and U-Net ranging from 0.09% to 20.05%, using all four methods used to determine accuracy (Dice coefficient, intersection over union, sensitivity, and accuracy). As shown in Figure 5(aii), the MAP sample was manually segmented (Figure 5(aiii)) without following the uniform standard. The segmentation based on FCN, U-Net, and HyNet was binary (white) compared with under-segmented (red) and over-segmented (blue) results. The under-segmented HyNet was the best solution.

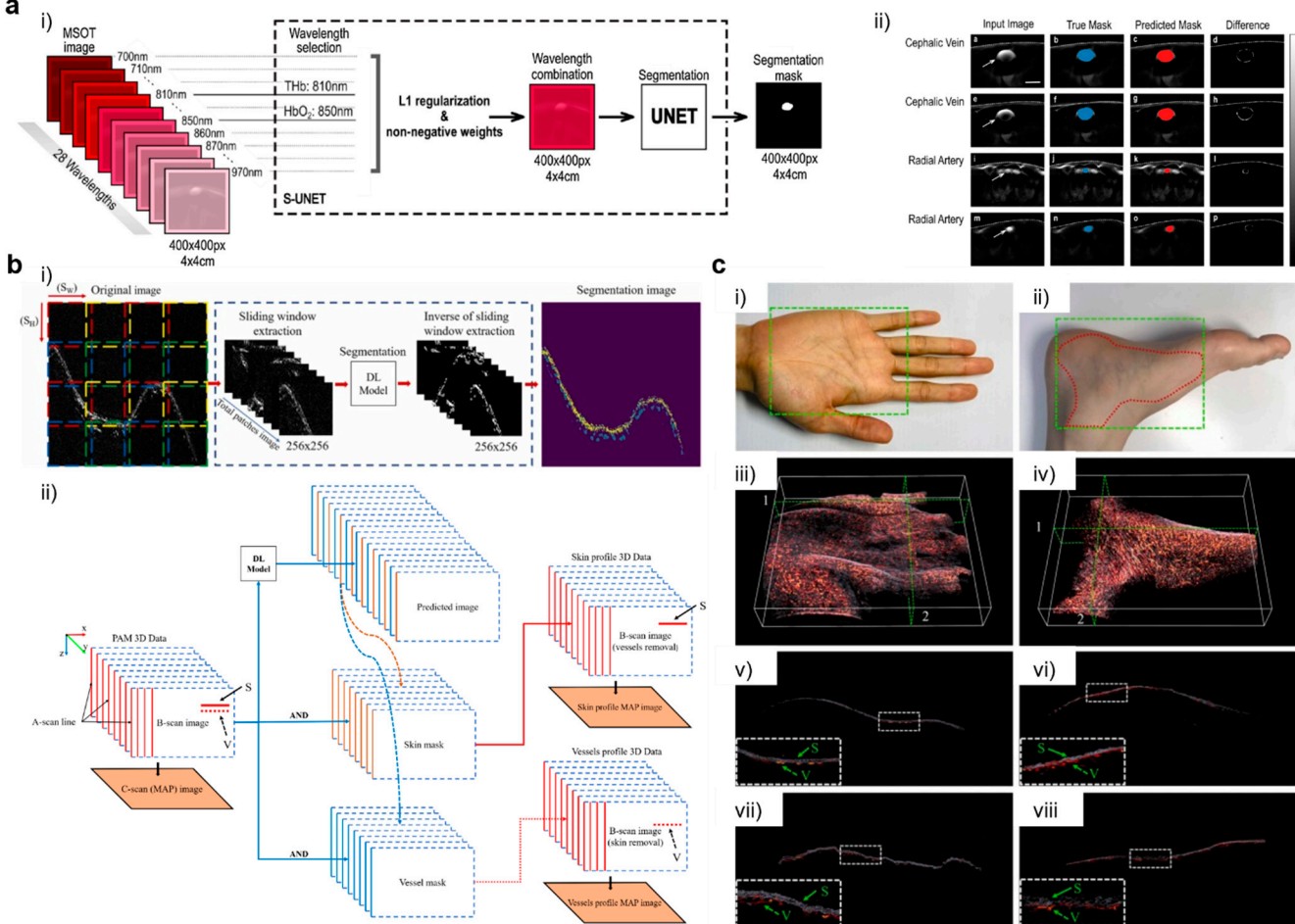

**Figure 4.** (**a**) (i) The S-UNET workflow identified illumination wavelengths; (ii) the results of segmented human vasculature based on an MSOT image in a sample 850 nm channel (input image). Compared with the ground truth in blue areas (true mask), the S-UNET showed the predicted segmentation masks in red areas (predict mask), and the absolute difference between them showed almost a complete overlap (Difference). Reproduced with permission from Chlis et al. [110]; published by Elsevier, 2021; (**b**) (i) The block diagram of processing used to transform high-resolution input image into a probability map. (ii) Formation principles of skin S and blood vessel V segmentation in a 3D volumetric image; (**c**) segmentation mapping of human skin and vascular system in (i, iii) human palm, (ii, iv) foot, and 3D PA volumetric image inside with marked ROIs (green). (v) Coronal plane of the (iii) human palm, (vii) sagittal plane of the (iii) human palm, (vi) coronal plane of the (iv) foot, (viii) sagittal plane of the (iv) foot. Reproduced with permission from Ly et al. [56]; published by Elsevier, 2021.

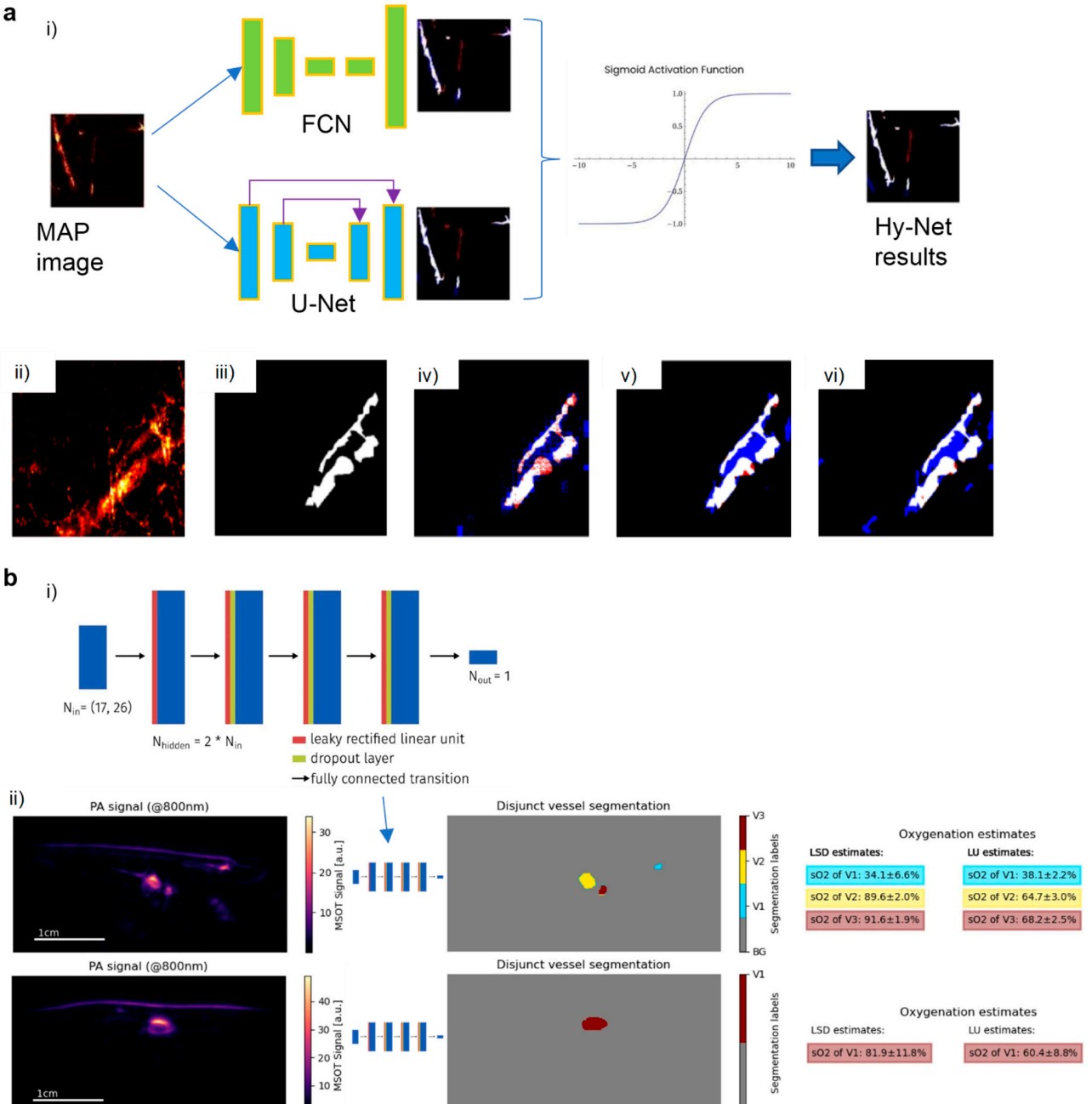

**Figure 5.** (**a**) (i) The structure of the Hy-Net-based concatenation of FCN and U-Net block followed by sigmoid activation; (ii–vi) visualization of segmentation: MAP images of sample, manual masks, FCN, U-Net, Hy-Net. Reproduced with permission from Yuan et al. [111]; published by OSA, 2020; (**b**) (i) visualization of the fully connected feed-forward neural network in LSD, (ii) Ii vivo results of LSD involving human forearms in B-scans and disjunct vessel segmentation. Reproduced with permission from Gröhl et al. [112]; published by *Nature*, 2021.

Another optoacoustic imaging was used to evaluate oxygenation by MSOT with a laser at wavelengths less than 1000 nm [42,117]. Blood oxygenation $sO_2$ was defined as the ratio of $HbO_2$ by total hemoglobin concentration $Hb = HbO_2 + HbR$ ($HbR$ : deoxygenated hemoglobin) based on image coordinates $(x, y)$ determined using Equation (6).

$$sO_2(x, y) = \frac{HbO_{2(x,y)}}{HbO_2(x, y) + HbR(x, y)} \times (\%) \tag{6}$$

The oxygen concentration was measured in $HbO_2$ and $HbR$ or $Hb$, and traditional methods (linear unmixing [118]) or supervised learning method (S-UNET [110]) were used, which required at least 02 wavelengths (linear unmixing) or predicted before S-UNET to solve the quantitative problem. Gröhl et al. [112] introduced the learned spectral decoloring (LSD) process in MSOT using an unsupervised learning method. The training dataset was generated via three simulations under different scenarios (generic tissue without skin specificity, geometry of oxygenation flow, and mimicking all features of human forearm structures). After training ground-truth with the fully connected feed-forward neural network models to explain domain-specific differences (Figure 5(bi)), the architectures were used as a baseline study using a single-pixel approach without CNNs. As shown in Figure 5(bii), the effect of learned spectral decoloring human forearms showed disjunct vessels using the phantom trained model. Based on the B-scan image at 800 nm wavelength, the model showed disjunct blood oxygenation levels of the arterial vessel structure compared with the results showing that linear unmixing (up to 68%) was a lower $sO_2$ estimate than LSD (up to 91%). Luke et al. [113] estimated blood oxygenation via spectroscopic photoacoustic (sPA) imaging using O-Net to segment the vessels from surrounding background tissues in less than 50 ms with a 5.1% median error after training with a three-dimensional Monte Carlo simulation dataset on breast tissue.

## 3. Discussion

Compared with traditional modalities such as CT, MRI, PET, and US, PAI is a non-invasive biomedical imaging technique, albeit with limited application in the clinical imaging field. Current quantitative and segmentation techniques involving PAI are improving its practical application in modern healthcare system [119–121]. All methods of classical segmentation were referenced and validated for experimental and practical use in PAI. Typically, using morphological segmentation methods, the microvascular structure was mapped and visualized via the microscale and multi-layer blood mapping of surface. The wavelet-based frequency segmentation methods were used to segment the tumor cells or oxygenated, and deoxygenated blood levels based on molar excitation coefficients. The simplest segmentation method using thresholding cleaned the PAI image in the preprocessing step to define the severity of breast cancer based on BI-RADS category to prepare for deep learning methods. Each mathematical method is only used to measure each factor, such as the intensity level changes (wavelet-based frequency, thresholding), diameter (thresholding, morphological), and tortuosity (thresholding, morphological), of the tissues. To measure all factors with an uncomplicated process, deep learning is the needed solution (Hy-Net, O-Net, etc.).

The deep learning technique used in the last five years can be used to overcome the system limitations and will continue to be researched as proposed to improve the quantitative segmentation of PAI. As an alternative to classical segmentation and quantitative methods, deep learning was utilized based on the features and properties of tumors or vessels in PAI. The knowledge could be used in supervised or unsupervised learning. Along with development, deep learning also enables the publication and extension of knowledge using open sources [54]. Many competitions or challenges each year often require solutions and cross-checking between participants to strive for the best results as the meaning of "learning".

## 4. Conclusions

QPAI is important to accurately obtain quantitative estimates of concentration and physiologically interesting chromophores. Based on segmentation, PAI systems provide easy access for visualization of human body structure and functionality. Classical PAI segmentation algorithms are readily available in modern systems. Due to rapid advances, deep learning's ideals have become the fastest tools that can be used to exploit all best advantages for clinical application. One of the key advantages of deep learning is the extremely fast responding time with minimized hardware configuration, which is hoped to enable real-time processing of the PAI system. Given the current clinical application and

standardization of PAI, the scale of clinical data will be increased but still limited by legal and human rights issues [122]. Therefore, the classical segmentation approaches will still be developed, as well as be applied in the PAI system. Further studies investigating the potential role of deep learning strategies in PAI are needed before clinical application can be envisaged.

**Author Contributions:** Conceptualization, T.D.L., S.-Y.K. and C.L.; resources, T.D.L. and C.L.; writing—original draft preparation, T.D.L., S.-Y.K. and C.L.; writing—review and editing, C.L.; visualization, T.D.L. and C.L.; project administration, C.L.; funding acquisition, C.L. All authors have read and agreed to the published version of the manuscript.

**Funding:** NRF grant funded by the Korean government (MSIT) (NRF-2019R1F1A1062948) and Bio 501 & Medical Technology Development Program of the NRF funded by the Korean government (MSIT) 502 (NRF-2019M3E5D1A02067958).

**Conflicts of Interest:** The authors declare no conflict of interest.

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
