# Peer review of "Segmentation and Quantitative Analysis of Photoacoustic Imaging: A Review"

_photonics, doi:10.3390/photonics9030176_

Round 1

Reviewer 1 Report

Accurate image segmentation has long been an obstacle to the clinical translation of PAI. This work summarizes the current segmentation and quantitative analysis in PAI. The classic approaches and the newly developed deep learning network architectures are all covered in this manuscript. It is an important step in the clinical translation of PA technology. Overall, the manuscript was well organized and carefully written. It fits well into the journal's scope. The work can be recommended for publication in Photonics.

Author Response

Thank you. 

Reviewer 2 Report

This paper reviews segmentation and quantitative analysis methods used in PA imaging for both classical and deep learning-based techniques. Overall, the paper is well-written and I recommend this paper to be published with a minor revision for some comments below.

  1. I recommend authors to further emphasize why this review is significant from the abstract. I can see the purpose of segmentation “to directly calculate the diameter, density, and tortuosity of the tissues”, but it would be clearer to describe why we need that information by segmentation.
  2. From the line 72 “One of the key advantages of deep learning is the extremely fast response that enables real-time processing of the PA system”, is this a general statement? It may be better to have an example or a requirement for this statement.
  3. Table 1 presents three types of segmentation methods, but I am wondering if there is a reason excluding other classic segmentation methods, including region-growing based and active contour-based methods.
  4. In the line 256, would you clarify what is a certain delay in the statement? Would this refer a segmentation processing time?
  5. The second paragraph in 2.2.2 seems too long. It may be better dividing it into two for improved readability. It looks many deep learning methods are just arrayed.
  6. What are the overall research trends moving on with this subject? I don’t clearly see what pros and cons of each type of segmentation method are. Would you discuss this point in the discussion? Specifically, you can discuss what is the general direction of this research area in the current and in the future and what are the points the researchers need to have
    in their mind.
  7. In the line 8, “is a emerging” has to be “is an emerging”

Author Response

Thank you for your fruitful comments. 
